# Progressive Sarcopenia Correlates with Poor Response and Outcome to Immune Checkpoint Inhibitor Therapy

**DOI:** 10.3390/jcm10071361

**Published:** 2021-03-25

**Authors:** Sven H. Loosen, Vincent van den Bosch, Joao Gorgulho, Maximilian Schulze-Hagen, Jennis Kandler, Markus S. Jördens, Frank Tacke, Christina Loberg, Gerald Antoch, Tim Brümmendorf, Ulf P. Neumann, Christiane Kuhl, Tom Luedde, Christoph Roderburg

**Affiliations:** 1Clinic for Gastroenterology, Hepatology and Infectious Diseases, University Hospital Düsseldorf, Medical Faculty of Heinrich Heine University Düsseldorf, 40225 Düsseldorf, Germany; jennis.kandler@med.uni-duesseldorf.de (J.K.); markus.joerdens@med.uni-duesseldorf.de (M.S.J.); christoph.roderburg@med.uni-duesseldorf.de (C.R.); 2Department of Diagnostic and Interventional Radiology, University Hospital RWTH Aachen, Pauwelsstr. 30, 52074 Aachen, Germany; vvandenbosch@ukaachen.de (V.v.d.B.); mschulze@ukaachen.de (M.S.-H.); ckuhl@ukaachen.de (C.K.); 3Department of Oncology, Hematology and Bone Marrow Transplantation with Section of Pneumology, University Medical Centre Hamburg-Eppendorf, Martinistraße 52, 20251 Hamburg, Germany; j.gorgulho@uke.de; 4Department of Hepatology and Gastroenterology, Charité University Medicine Berlin, Augustenburger Platz 1, 13353 Berlin, Germany; frank.tacke@charite.de; 5Department of Diagnostic and Interventional Radiology, University Hospital Düsseldorf, Medical Faculty of Heinrich Heine University Düsseldorf, 40225 Düsseldorf, Germany; christina.loberg@med.uni-duesseldorf.de (C.L.); gerald.antoch@med.uni-duesseldorf.de (G.A.); 6Department of Medicine IV, University Hospital RWTH Aachen, Pauwelsstr. 30, 52074 Aachen, Germany; tbruemmendorf@ukaachen.de; 7Department of Visceral and Transplantation Surgery, University Hospital RWTH Aachen, Pauwelsstr. 30, 52074 Aachen, Germany; uneumann@ukaachen.de

**Keywords:** sarcopenia, checkpoint inhibitors, ICI, PD-L1, PD-1, prognosis, body composition

## Abstract

Background: Immune checkpoint inhibitors (ICIs) represent a new therapeutic standard for an increasing number of tumor entities. Nevertheless, individual response and outcome to ICI is very heterogeneous, and the identification of the ideal ICI candidate has remained one of the major issues. Sarcopenia and the progressive loss of muscle mass and strength, as well as muscular fat deposition, have been established as negative prognostic factors for a variety of diseases, but their role in the context of ICI therapy is not fully understood. Here, we have evaluated skeletal muscle composition as a novel prognostic marker in patients undergoing ICI therapy for solid malignancies. Methods: We analyzed patients with metastasized cancers receiving ICI therapy according to the recommendation of the specific tumor board. Routine CT scans before treatment initialization and during ICI therapy were used to assess the skeletal muscle index (L3SMI) as well as the mean skeletal muscle attenuation (MMA) in *n* = 88 patients receiving ICI therapy. Results: While baseline L3SMI and MMA values were unsuitable for predicting the individual response and outcome to ICI therapy, longitudinal changes of the L3SMI and MMA (∆L3SMI, ∆MMA) during ICI therapy turned out to be a relevant marker of therapy response and overall survival. Patients who responded to ICI therapy at three months had a significantly higher ∆L3SMI compared to non-responders (−3.20 mm^2^/cm vs. 1.73 mm^2^/cm, *p* = 0.002). Moreover, overall survival (OS) was significantly lower in patients who had a strongly decreasing ∆L3SMI (<−6.18 mm^2^/cm) or a strongly decreasing ∆MMA (<−0.4 mm^2^/cm) during the first three month of ICI therapy. Median OS was only 127 days in patients with a ∆L3SMI of below −6.18 mm^2^/cm, compared to 547 days in patients with only mildly decreasing or even increasing ∆L3SMI values (*p* < 0.001). Conclusion: Both progressive sarcopenia and an increasing skeletal muscle fat deposition are associated with poor response and outcome to ICI therapy, which might help to guide treatment decisions during ICI therapy.

## 1. Introduction

The introduction of immune checkpoint inhibitors (ICIs) into therapeutic management of patients with malignant tumors has changed our view on how to treat cancer [1]. Due to the superior efficacy and safety, ICI’s represent the standard treatment for an increasing number of cancer entities. By directly targeting so-called immune checkpoints (e.g., PD-L1/PD-1 or B7/CTLA-4), ICIs activate the immune system to induce tumor cell death. In recent years, it has become clear that only a subset of patients exhibit durable tumor responses to ICI therapy, and the concept of so called “cold” and “hot” tumors has been developed. “Hot tumors” are characterized by an already existing adaptive immune response with CD8+ T cell infiltration, IFN-γ signaling, and efficient presentation of tumor antigens. Checkpoint blockade then activates this pre-existing response, leading to tumor response and increased patient survival.

Besides tumor response, several other factors are of prognostic relevance in patients with cancer. As such, body composition was identified as a potential factor determining overall survival of patients with cancer. It turned out that low skeletal muscle mass and sarcopenia are associated with an impaired prognosis in patients suffering from malignancies (e.g., [2,3]). Sarcopenia is defined as “*the progressive loss of muscle mass and strength with a risk of adverse outcomes such as disability, poor quality of life and death*” by the Special Interest Group of the European Sarcopenia Working Group in 2010 [4]. Interestingly, in many cancers, sarcopenia is not only a predictor of survival but also a predictor of chemotoxicity in patients with cancer receiving platinum and taxane-based chemotherapy and might affect outcome in patients receiving systemic treatments [5,6,7]. In contrast, the role of sarcopenia as a prognostic marker in patients treated with ICI has not been investigated so far.

In this study, we examined whether sarcopenia evaluated by the skeletal muscle index (L3SMI) or myosteatosis evaluated by the median skeletal muscle attenuation (MMA) might also represent a prognostic factor in patients receiving ICI.

## 2. Patients and Methods

### 2.1. Study Design and Patient Characteristics

The present study aims to evaluate the predictive and/or prognostic role of skeletal muscle composition in patients undergoing immune checkpoint inhibitor (ICI) therapy for solid malignancies. Eighty-eight patients scheduled to receive ICI at the interdisciplinary cancer outpatient clinic at University Hospital RWTH Aachen were recruited between 2017 and 2020 (see Table 1 for patient characteristics). The study protocol was approved by the ethics committee of the University Hospital RWTH Aachen, Germany (EK 206/09) and conducted in accordance with the ethical standards laid down in the Declaration of Helsinki. Written informed consent was obtained from the patients.

### 2.2. Assessment of Body Composition and Definition of the Skeletal Muscle Index (L3SMI)

Assessment of the skeletal muscle composition was based on CT scans in the venous phase with a slice thickness of 1 mm. CT scans before and at three months after treatment initialization were evaluated. Total skeletal muscle area and median skeletal muscle attenuation (MMA) were manually assessed on axial CT scans at the center plane of the 3rd lumbar vertebra (LV) using the semi-automatically segmentation tool “3D slicer” [8]. The following muscle groups were segmented: rectus abdominis, external and internal obliques, transversus abdominis, and quadratus lumborum, as well as the psoas major and erector spinae (Figure 1). Muscles were quantified by using attenuation values between −29 to 150 Hounsfield Units (HU). The MMA was calculated by the software automatically. We normalized the skeletal muscle area for the patients’ height and defined the skeletal muscle index (SMI):
(1)$$Skeletal\;muscle\;index\;\left(L3SMI\right)\;\left[mm^{2}/cm\right]=\;\frac{skeletal\;muscle\;area\;at\;3rd\;LV\;\left[mm^{2}\right]}{body\;height\;\left[cm\right]}$$

### 2.3. Evaluation of Tumor Response and Outcome

The patients’ individual tumor response to ICIs was evaluated using cross-sectional imaging modalities (CT or MRI scan) three months after treatment initialization based on iRECIST criteria, when applicable [10]. Tumor response was classified using the standard nomenclature for RECIST: Complete response (CR), partial response (PR), stable disease (SD), and progressive disease (PD). CR, PR, and SD were defined as “disease control” (DC), whereas patients with PD were classified into non-DC. Overall survival (OS) was defined as the time period between the first administration of ICI and death. The median time of follow-up (time between first administration of ICI therapy and death/censoring) was 257 days (IQR: 127–526).

### 2.4. Statistical Analysis

The Shapiro–Wilk test was performed to test for normal distribution. The Mann–Whitney-U test and Kruskal–Wallis test were used to compare non-parametric data. The Wilcoxon signed-rank test was used to compare related samples. Box plots display a graphical summary of the median, quartiles, and ranges. The Spearman correlation coefficient was used for correlation analyses. Kaplan–Meier curves were used to evaluate the impact of L3SMI and MMA on the overall survival (OS), and the log-rank test was used to test for statistical differences between two subgroups. The optimal prognostic cut-off value was calculated by fitting Cox proportional hazard models to the dichotomized survival status and survival time and then defining the optimal cut-off as the point with the most significant split in log-rank test. The prognostic value of variables was further tested in uni- and multivariate Cox-regression analyses. Parameters with a *p*-value <0.100 in univariate analyses were included into the multivariate Cox-regression analysis. The hazard ratio (HR) and the 95% confidence interval are displayed. All statistical analyses were performed with SPSS 23 (SPSS, Chicago, IL, USA) and RStudio 1.2.5033 (RStudio Inc., Boston, MA, USA) [11]. A *p*-value of <0.05 was considered statistically significant (* *p* < 0.05; ** *p* < 0.01; *** *p* < 0.001).

## 3. Results

### 3.1. Patient Baseline Characteristics

The median age of the cohort was 67 years (range: 34–87 years), and 65.9% of patients were male. Of the patients, 39.8% were treated with ICI for non-small cell lung cancer (NSCLC), 15.9% for malignant melanoma, 13.6% for urothelial carcinoma, 13.6% for gastrointestinal (GI) cancer, 8.0% for head and neck cancer, and 9.1% for other malignancies. The majority of patients (92.9%) presented with Union for International Cancer Control (UICC) IV tumor stage. In terms of response to ICI therapy, 46.6% of patients had a controlled disease (DC) at three months. During the follow-up period, 63.6% of patients died. Table 1 summarizes the cohort characteristics.

### 3.2. Assessment of Skeletal Muscle Composition in Patients Receiving ICI Therapy

We defined the skeletal muscle index (L3SMI) as a surrogate for the total amount of skeletal muscle in order to identify patients with sarcopenia (see Patients and Methods for details). The median baseline L3SMI before treatment initiation was 76.79 mm^2^/cm (range: 45.99–124.91 mm^2^/cm) and did not significantly differ between tumor entities (Appendix A
Appendix A), UICC tumor stage (Appendix A
Appendix A), or patients with different Eastern Cooperative Oncology Group (ECOG) performance status (PS) (Appendix A
Appendix A). Male patients, however, had a significantly higher L3SMI compared to female patients (Appendix A
Appendix A). Moreover, we did not observe a significant difference of the L3SMI between patients who had previously received systemic chemotherapy (Appendix A
Appendix A). The L3SMI did not correlate with patients’ age (r_S_: 0.066, *p* = 0.546).

As the second parameter of the patients’ body composition, we next determined the median skeletal muscle attenuation (MMA) as a surrogate parameter for muscular fat deposition. The median MMA was 34.55 Hounsfield units (HU) (range: 16.4–51.0 HU) and again did not significantly differ between tumor entities (Appendix A
Appendix A). Interestingly, we observed a strong trend towards a lower MMA (*p* = 0.073), suggesting a higher amount of muscular fat deposition in patients with more advanced tumor stage (Appendix A
Appendix A) as well as a significant stepwise reduction of the MMA in patients with an increasingly impaired ECOG PS (Appendix A
Appendix A). On the contrary, male and female patients had comparable MMA values (Appendix A
Appendix A). Interestingly, muscular fat deposition was significantly more pronounced (lower MMA) in patients who previously received systemic chemotherapy before initiation of ICI therapy (Appendix A
Appendix A). The MMA negatively correlated with patients’ age (r_S_: −0.318, *p* = 0.003).

### 3.3. Progressive Sarcopenia Is Associated with a Poor Response to ICI Therapy

We subsequently evaluated whether baseline L3SMI and MMA might be predictive for the patients’ individual response to ICI therapy and divided our cohort into patients who did or did not respond to therapy in terms of a disease control (DC) at three months after ICI treatment initialization (DC = 41, non-DC = 47). However, DC and non-DC patients revealed a comparable initial L3SMI and MMA (Figure 2A,B). In the next step, we hypothesized that the individual longitudinal course of the L3SMI and MMA, rather than the baseline value, might reflect treatment response to ICI therapy, and we evaluated the patients’ body composition three months after therapy initialization to identify patients with increasing or decreasing L3SMI/MMA values (delta (∆)L3SMI and ∆MMA). The median L3SMI and MMA value was not significantly altered between the baseline CT and at 3 months (Figure 2C,D). In total, we identified *n* = 39 and *n* = 40 patients with increasing as well as *n* = 41 and *n* = 40 patients with decreasing L3SMI and MMA, respectively. Interestingly, patients with a controlled disease at three months had a significantly higher median ∆L3SMI (−3.20 mm^2^/cm vs. 1.73 mm^2^/cm, *p* = 0.002) and a strong trend towards a higher median ∆MMA (−1.0 HU vs. 0.89 HU, *p* = 0.090) compared to non-DC patients (Figure 2E,F), meaning that patients with increasing sarcopenia or muscular fat deposition were less likely to be responders to ICI therapy. In line with this, binary logistic regression analysis revealed ∆L3SMI as a significant predictive marker for treatment response at three months (OR: 1.127 [1.044–1.216], *p* = 0.002).

### 3.4. Progressive Sarcopenia and Myosteatosis Are a Negative Prognostic Factors For ICI Therapy

Next, we aimed to evaluate whether the baseline skeletal muscle composition or its longitudinal alteration during the course of ICI therapy might also have a prognostic relevance in terms of overall survival (OS). Therefore, we first compared the OS between patients with high or low L3SMI/MMA in the baseline CT scan using the 50th percentile as a cut-off value. Here, we observed only a slight trend towards a better outcome in patients with L3SMI/MMA values above the cohort’s median (76.79 mm^2^/cm and 34.55 HU, Figure 3A and B). In a next step, we established ideal prognostic cut-off values for the L3SMI and MMA, which best discriminated between patients with impaired outcomes and long-term survivors. However, despite the clear trend towards a poor outcome in patients with low L3SMI and MMA values, statistical significance was still not reached (*p* = 0.133 and *p* = 0.196, Figure 3C,D).

Hypothesizing that the longitudinal alteration of the L3SMI and MMA rather than the patients’ baseline muscle composition might represent the better prognostic marker, we subsequently compared the OS between patients with increasing or decreasing L3SMI/MMA between baseline CT and at three months after ICI therapy initialization. Strikingly, we observed that patients with a positive ∆L3SMI had a significantly better OS compared to patients with a negative ∆L3SMI (Figure 4A). In line with this, a negative ∆MMA was associated with a strong trend towards an impaired OS (*p* = 0.069, Figure 4B). In a next step, we established ideal prognostic L3SMI- and MMA-cut-off values that best discriminates between long-term survivors and patients who died early during ICI therapy. When applying these cut-off values, Kaplan-Meier curve estimates revealed that both a strongly decreasing ∆L3SMI (<−6.18 mm^2^/cm) as well as a strongly decreasing ∆MMA (<−0.4 mm^2^/cm) during the first three month of ICI therapy were associated with a significantly worse outcome (Figure 4C,D). As such, patients with a ∆L3SMI of below −6.18 mm^2^/cm had a median OS of only 127 days in contrast to patients with an only mildly decreasing or even increasing ∆L3SMI who had a 4.3-times higher median OS of 547 days.

Finally, we used uni- and multivariate Cox-regression analyses to further corroborate the prognostic relevance of progressive sarcopenia and myosteatosis (Table 2). In univariate analysis, ∆L3SMI turned out as a prognostic factor for OS (HR: 0.929, 95% CI: 0.898–0.960, *p* < 0.001). Univariate Cox-regression analysis further revealed a HR of 0.940 (95% CI: 0.881–1.003, *p* = 0.062) for ∆MMA. Importantly, the prognostic relevance of ∆L3SMI was independent of several potentially confounding variables including tumor stage, ECOG PS, BMI and laboratory markers of organ dysfunction in multivariate Cox-regression analysis (HR: 0.925, 95% CI: 0.890–0.961, *p* < 0.001). In line with this, progressive myosteatosis, in terms of a ∆MMA below −0.4 mm^2^/cm turned out to be a significant predictor of OS in univariate Cox-regression analysis (HR: 0.523, 95% CI: 0.300–0.913, *p* = 0.023).

## 4. Discussion

In the present study, we analyzed the impact of sarcopenia as a prognostic and predictive marker in patients receiving ICI therapy for cancer. We have demonstrated that both progressive sarcopenia as well as an increasing skeletal muscle fat deposition, both assessed as early as three months after treatment initiation, are associated with a poor response and outcome to ICI therapy. These findings, which were independent from the tumor entity and the ICI compound, might help to guide therapy decisions in cancer patients and to identify patients that particularly benefit from ICI therapy in terms of overall survival.

In recent years, ICIs have been approved for the treatment of an increasing number of tumor entities. Many of the pivotal trials demonstrated superior efficacy and lower toxicity of ICI compared to classical chemotherapy [12,13]. However, only a subset of patients benefits from ICI therapy, while others experience rapid tumor progression and poor prognosis [14]. In this context, great efforts have been made to identify markers that reliably predict tumor response and/or survival of patients receiving ICI therapy [15]. Most of the existing stratification systems rely on the analyses of tumor tissue. Most importantly, patients with microsatellite instability-high (MSI-H), or mismatch repair deficient (dMMR) malignancies show more frequent and deeper tumor response when compared to other patients [16,17,18]. Based on these data, the FDA approved the use of ICI for any solid MSI-H tumor. In addition to MSI, PD-L1 expression levels were recently established as a marker predicting response in patients with different cancer. Most recently, it was demonstrated that patients with PD-L1 positive gastric cancer show improved response rates and improved survival compared to patients without PD-L1 expression [19]. While the same was true, e.g., for patients with non-small cell lung cancer [20], PD-L1 tumor expression was neither predictive nor prognostic in other tumor entities such as hepatocellular carcinoma (HCC) [21]. In terms of circulating biomarkers for ICI therapy, data are scarce; some authors have suggested that serum lactate dehydrogenase (LDH) levels [22,23] and the neutrophil/lymphocyte ratio (NLR) might correlate with an unfavorable prognosis in patients with melanoma and/or NSCLC treated with ICI. Thus, easily accessible markers for the estimation of response and/or prognosis in the context of ICI are urgently needed.

To our best knowledge, we have demonstrated for the first time a potential prognostic and predictive role of individual body composition in the context of ICI treatment. Body composition can be analyzed using different methods, with bioelectrical impedance analysis or ultrasound representing the most frequently applied ones [24,25]. Because computed tomography images are available for almost all patients with cancer, we decided to access muscle mass (sarcopenia) and muscle quality (myosteatosis) by calculating the L3SMI and MMA, as described above. Specifically, we show that, while baseline L3SMI or MMA were not associated to the clinical patients´ outcome, those patients who responded to ICI therapy at three months had a significantly higher ∆L3SMI compared to non-responders. Moreover, overall survival (OS) was significantly lower in patients who had a strongly decreasing ∆L3SMI (<−6.18 mm^2^/cm) or a strongly decreasing ∆MMA (<−0.4 mm^2^/cm) during the first three month of ICI therapy. As a possible explanation, it should be taken into account that progressive sarcopenia is observed in many elderly and moribund patients. Importantly, the prognostic role of ∆L3SMI turned out to be independent of the patients’ ECOG PS and BMI. Furthermore, muscle wasting is observed in the clinical course of many cancers due to denervation, fasting, cardiac failure, and renal dysfunction [26], which might themselves limit patients’ prognosis. Nevertheless, in our analysis, L3SMI/MMA values were neither associated with the cancer etiology nor to the disease stage or the line of therapy. Recent data have linked sarcopenia to the presence of systemic inflammation and the activation of the immune system. Therefore, progredient loss of muscle mass and quality in the subgroup of patients with a poor response and outcome to ICI might reflect a chronically activated immune system, which in turn negatively influences the anti-tumoral effects of ICI. In line with this hypothesis, we have recently demonstrated that elevated levels of suPAR, a marker for systemic inflammation and infection, reflect an impaired tumor response and patient’s prognosis. Moreover, in recent studies, the neutrophil-to-lymphocyte ratio (NLR) turned out to be a negative predictive and/or prognostic markers for ICI therapy [27]. However, further molecular studies are warranted to fully dissect a potential functional role of muscle wasting and sarcopenia in the context of ICI therapy.

Despite the fact that we show for the first time a possible role of sarcopenia and myosteatosis in ICI, it is important to note that our study is limited by different factors. Most importantly, our study featured a so-called basket design, highlighting that we analyzed a heterogeneous cohort of patients suffering from various tumor entities. Although we did not find differences of the L3SMI and MMA between malignancies, arguing for an entity independent prognostic role of the body composition, further analyses are needed to provide a more complete picture of the role of sarcopenia and myosteatosis within the different tumor entities. Since we only examined patients treated with ICI but not patients receiving alternative treatments such as conventional chemotherapy, we cannot provide information on a treatment specific role of ICI. Moreover, our database did not feature information regarding ethnicities, which potentially might have an influence on body composition. In addition, there is some variation in the literature about methods of adjustments regarding the L3SMI [9], which should be taken into account when comparing results between studies. Finally, distinct comorbidities, including COPD (chronic obstructive pulmonary disease), that are more frequent in specific subgroup of patients might have biased the results. Interestingly, recent data has demonstrated that sarcopenia might be predictive for chemotoxicity and treatment outcome in patients with ovarian cancer, highlighting the need for more specific studies [5,6,7]. Thus, further clinical trials including larger, more clearly defined patient cohorts as well as molecular studies are needed to fully unravel the role of sarcopenia and myosteatosis in ICI-treated cancer patients. We hope that our exploratory study might give rise to such research.

## 5. Conclusions

Our data suggest a previously unrecognized role of sarcopenia and myosteatosis in cancer patients treated with ICI. Given that our data can be confirmed in prospective clinical trials, sarcopenia and myosteatosis might emerge as novel stratification tools in patients receiving ICI and complement existing algorithms to guide early treatment decisions in the context of ICI therapy.

## Figures and Tables

**Figure 1 jcm-10-01361-f001:**
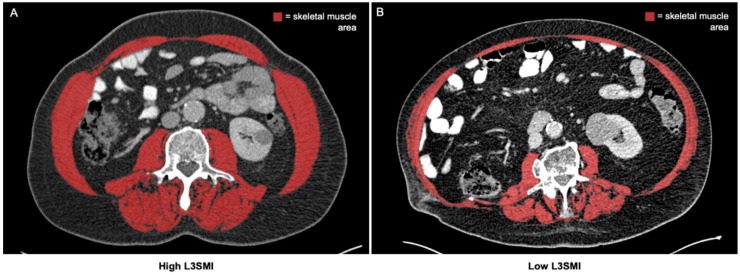
Determination of the total skeletal muscle area on axial CT scans. The total skeletal muscle area is assessed on axial CT scans at the center plane of the 3^rd^ lumbar vertebra and normalized for the patients’ height (L3SMI). (**A**) Exemplary CT scan of a patient with high L3SMI. (**B**) Exemplary CT scan of a patient with low L3SMI [9].

**Figure 2 jcm-10-01361-f002:**
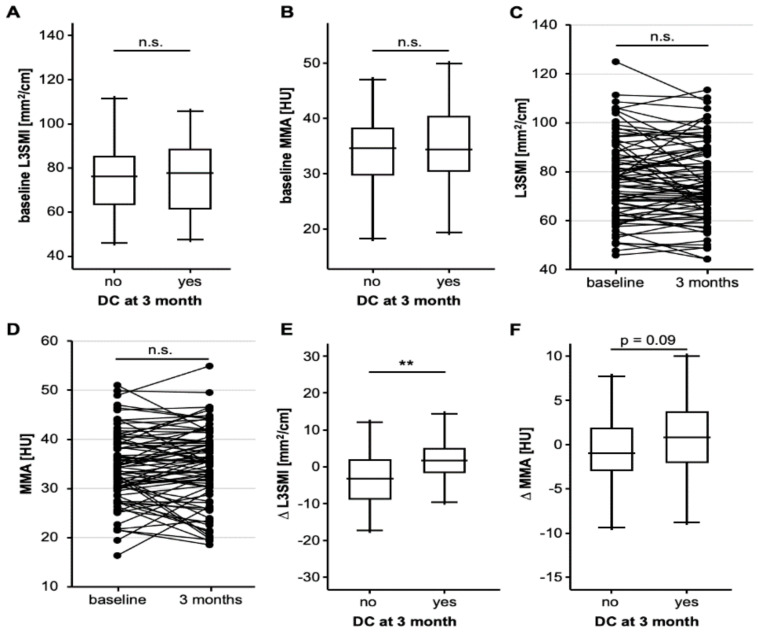
Progressive sarcopenia is associated with a poor response to ICI (immune check-point inhibitor) therapy (**A**,**B**) Baseline L3SMI and MMA values are comparable between patients who did or did not respond to ICI therapy at three months. (**C**,**D**) Overall longitudinal L3SMI/MMA values between baseline CT (computed tomography) scans and at three months are unaltered. (**E**) Patients who did not respond to ICI therapy had a significantly lower ∆L3SMI (L3 skeletal muscle index) compared to responders. (**F**) Patients who did not respond to ICI therapy had a strong trend towards a lower ∆MMA (mean muscle attenuation) compared to responding patients. n.s. non significant, ** *p* < 0.01.

**Figure 3 jcm-10-01361-f003:**
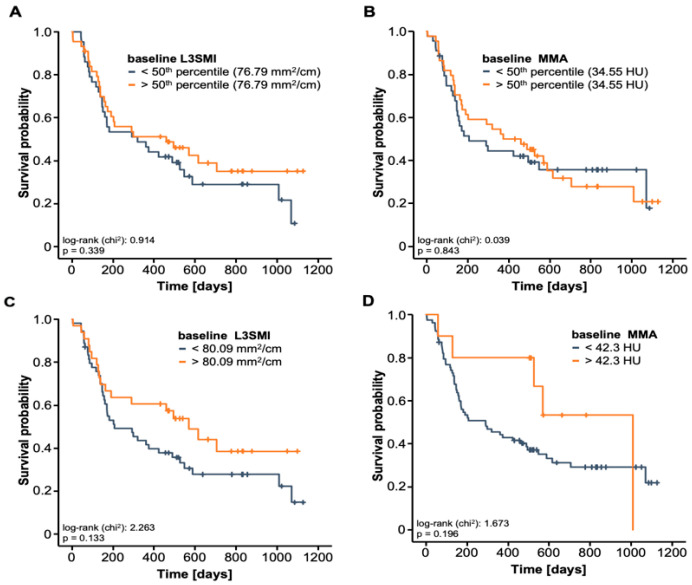
Baseline L3SMI and MMA values are unsuitable for predicting OS (**A**) Patients with L3SMI above the 50th percentile have a comparable OS compared to patients with L3SMI below this cut-off. (**B**) Patients with an initial MMA above the 50th percentile have a comparable OS compared to patients with L3SMI below this cut-off. (**C**) Using an ideal prognostic cut-off value (80.09 mm^2^/cm), patients with a baseline L3SMI below this cut-off show a trend towards an impaired OS. (**D**) Using an ideal prognostic cut-off value (42.3 HU), patients with a baseline MMA below this cut-off show a trend towards an impaired OS.

**Figure 4 jcm-10-01361-f004:**
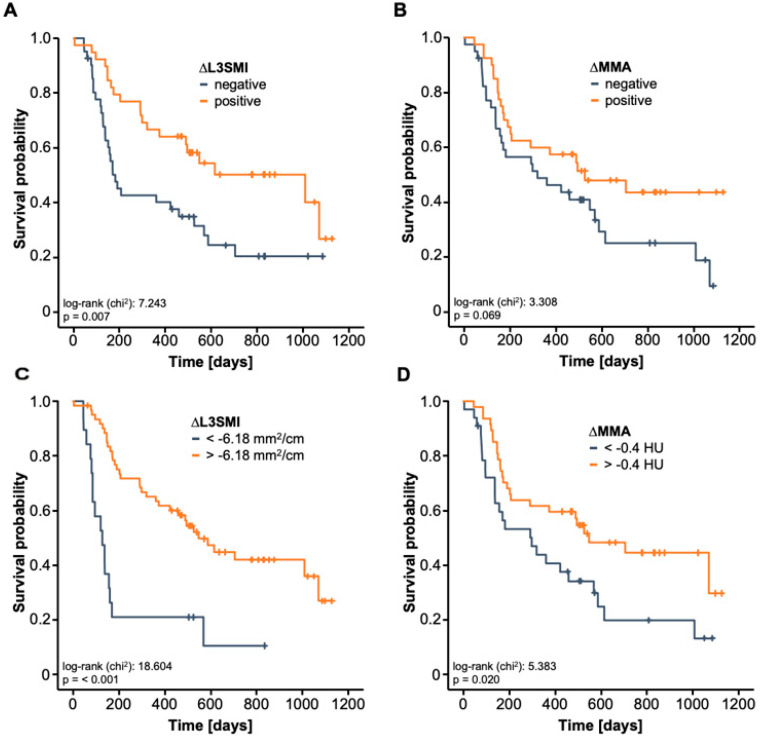
Deterioration of the skeletal muscle area and increasing muscular fat deposition are a negative prognostic marker for ICI therapy (**A**) Patient who reveal an increasing L3SMI value at 3 months (positive ∆L3SMI) have a significantly better OS compared to patients with a negative ∆L3SMI. (**B**) Patient who reveal an increasing MMA value at 3 months (positive ∆MMA) show a strong trend towards a better OS compared to patients with a negative ∆MMA. (**C**) A strongly decreasing ∆L3SMI (<−6.18 mm^2^/cm) is associated with a highly significantly reduced OS. (**D**) A strongly decreasing ∆MMA (<−0.4 mm^2^/cm) is associated with a significantly reduced OS.

**Table 1 jcm-10-01361-t001:** Patient characteristics.

Parameter	Study Cohort
Cancer patients	*n* = 88
L3SMI (mm^2^/cm, median and range) at baseline at 3 months	76.79 (45.99–124.91) 74.02 (44.23–113.47)
MMA (HU, median and range) at baseline at 3 months	34.55 (16.4–51.0) 35.4 (18.5–54.9)
Gender (%): male female	65.9 34.1
Age (years, median and range)	67 (34–87)
BMI (kg/m^2^, median and range)	24.5 (15.9–42.3)
Tumor localization (%): NSCLC Malignant melanoma Urothelial cancer GI cancer Head and neck cancer Others	39.8 15.9 13.6 13.6 8.0 9.1
Staging (%): UICC III UICC IV	7.1 92.9
ICI regimen (%): Nivolumab Pembrolizumab Nivolumab + Ipilimumab Others (e.g., Avelumab, Durvalumab)	55.7 28.4 9.1 6.8
Previous systemic therapy before ICI? (%): Yes No	69.3 30.7
ECOG PS (%): ECOG 0 ECOG 1 ECOG 2	9.1 52.3 38.6
Disease control at 3 months? (%): Yes No	46.6 53.4
Deceased during follow-up? (%): Yes No	63.6 36.4

L3SMI: L3 skeletal muscle index, MMA: mean muscle attenuation, BMI: body mass index, NSCLC: non-small cell lung cancer, GI: gastrointestinal, UICC: Union for International Cancer Control, ICI: immune checkpoint inhibitor, ECOG PS: Eastern Cooperative Oncology Group performance status.

**Table 2 jcm-10-01361-t002:** Uni- and multivariate Cox-regression analyses for the prediction of overall survival.

	Univariate Cox-Regression		Multivariate Cox-Regression	
Parameter	*p*-Value	Hazard-Ratio (95% CI)	*p*-Value	Hazard-Ratio (95% CI)
∆L3SMI	<0.001	0.929 (0.898–0.960)	<0.001	0.925 (0.890–0.961)
Age	0.999	1.000 (0.975–1.026)		
Sex	0.558	0.850 (0.495–1.462)		
BMI	0.024	0.939 (0.889–0.992)	0.110	0.952 (0.896–1.011)
UICC tumor stage	0.083	5.757 (0.794–41.726)	0.175	3.986 (0.540–29.417)
ECOG PS	0.050	1.533 (1.000–2.352)	0.032	1.717 (1.046–2.819)
Leukocyte count	0.670	1.013 (0.955–1.074)		
Sodium	0.335	0.969 (0.908–1.033)		
Potassium	0.568	0.860 (0.512–1.443)		
AST	0.357	1.004 (0.995–1.014)		
Bilirubin	0.064	1.688 (0.970–2.938)	0.747	0.810 (0.224–2.927)
Creatinine	0.629	1.149 (0.653–2.023)		
LDH	0.924	1.000 (0.998–1.002)		

BMI: Body-Mass-Index, UICC: Union for international cancer control, AST: aspartate transaminase, ECOG PS: Eastern Cooperative Oncology Group performance status, LDH: lactase dehydrogenase.

## Data Availability

All data are available on reasonable request from the corresponding author.

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
