# Peer review of "Progressive Sarcopenia Correlates with Poor Response and Outcome to Immune Checkpoint Inhibitor Therapy"

_jcm, 2021, doi:10.3390/jcm10071361_

Round 1
Reviewer 1 Report
In the present study, the authors analyzes the impact of sarcopenia as a prognostic and predictive marker in patients receiving ICI therapy for cancer.
The authors demonstrate that progressive sarcopenia and an increasing skeletal muscle fat deposition are associated with a poor response and outcome to ICI therapy
Overall, it is a good study. It addresses an important question, relevant and important in the field of solid tumors treatment.
- A conclusion section is missing and should added
Author Response
In the present study, the authors analyzes the impact of sarcopenia as a prognostic and predictive marker in patients receiving ICI therapy for cancer. The authors demonstrate that progressive sarcopenia and an increasing skeletal muscle fat deposition are associated with a poor response and outcome to ICI therapy. Overall, it is a good study. It addresses an important question, relevant and important in the field of solid tumors treatment. A conclusion section is missing and should added.
We are grateful to the reviewer for her/his constructive and highly positive evaluation of our manuscript. In the revised version of the manuscript, we now provide a “conclusion section” as kindly suggested:
“Our data suggest a previously unrecognized role of sarcopenia and myosteatosis in cancer patients treated with ICI. Given that our data can be confirmed in prospective clinical trials, sarcopenia and myosteatosis might emerge as novel stratification tools in patients receiving ICI and complement existing algorithms to guide early treatment decisions in the context of ICI therapy.” (page 10 of the revised manuscript).
Reviewer 2 Report
I enjoyed reading this manuscript and thought it was a well conducted study with some interesting results. I have mainly minor comments to make which I think will improve the manuscript as a whole. A more general comment is that throughout the manuscript the term sarcopenia is used to describe muscle loss. This is misleading as muscle strength or performance were never measured. I suggest this needs to be changed throughout and some comment made in either the introduction or discussion about how muscle mass is related to sarcopenia.
Abstract
There is no mention of which group of patients were receiving the ICI therapy as this is not a cancer journal specifically I think this probably needs clarifying.
Keywords
The keywords mention cachexia but not sarcopenia. Cachexia is mentioned nowhere within the manuscript.
Introduction
There is a poorly defined and increasingly murky relationship between cachexia and sarcopenia. There is likely to be a large degree of overlap and/or, both conditions being syndromes within the same spectrum of pathophysiology. However, cachexia is an accepted term when considering muscle loss in cancer and therefore it should be introduced within the background information with a brief discussion about terminology. In particular considering why sarcopenia has been chosen as a term for this study especially given that measures of muscle strength and performance have not been included.
Methods
L3SMI formula appears to be referenced wrong.
There is some variation in the literature about whether L3SMI includes adjustment for height or height squared. Please reference appropriately.
I can not comment on methodology pertaining to tumour response.
I was unable to find the length of follow up in the methodology which is important given the quotation of survival statistics later on.
I would also mention the multivariable analysis within the statistical methods.
Results
There is no mention of ethnicities in the patient characteristics. This can have a significant impact on body composition. This needs to be either included or considered as a limitation.
I would also like to see baseline L3SMI and MMA analysed with respect to age as this is an important factor within the development of sarcopenia.
You state that you identified patients with increasing and decreasing L3SMI and MMA. Reviewing the figure most of the patients appear to be static so the actual change would be very small and of little clinical relevance. I am not really sure this statement adds anything to the manuscript at all. I would probably remove it or clarify why you think it is important.
I would like to see the full univariable Cox analysis for MMA. Maybe in the supplementary data.
Discussion
Air displacement plethysmography is not a frequently used assessment tool of body composition. I would consider ultrasound if you want to quote another example.
Line 303 - grammatical error
Line 314 - no reference
I would really emphasise the survival data in the discussion eg the difference between 127 and 547 days. This is huge and clinically very important both for clinicians and patients. In my opinion what this paper has highlighted is 1) an excellent prognostic tool and 2) possible mechanisms for sarcopenia/muscle loss. Although point 2 is discussed I would try and emphasise both these points more in the discussion.
As a limitation to the methodology I would include the liklihood of concurrent multi-morbidity that could increase prevalence of sarcopenia with certain cancers. Eg NSCLC and COPD both of which can cause cachexia and have independent inflammatory effects.
Author Response
I enjoyed reading this manuscript and thought it was a well conducted study with some interesting results. I have mainly minor comments to make which I think will improve the manuscript as a whole. A more general comment is that throughout the manuscript the term sarcopenia is used to describe muscle loss. This is misleading as muscle strength or performance were never measured. I suggest this needs to be changed throughout and some comment made in either the introduction or discussion about how muscle mass is related to sarcopenia.
We want to thank the reviewer for her/his positive and constructive evaluation of our manuscript. Based on the reviewers´ comments, we carefully overworked the manuscript to provide all required clarifications.
Abstract
There is no mention of which group of patients were receiving the ICI therapy as this is not a cancer journal specifically I think this probably needs clarifying.
We fully agree with the reviewer that this information should be provided given the fact that many readers of the Journal of Clinical Medicine, covering the full spectrum medicine, will not be experts for cancer treatment. Please see the revised version of the abstract:
“We analyzed patients with metastasized cancers receiving ICI therapy according to the recommendation of the specific tumor board.” (page 1 of the revised manuscript).
Keywords
The keywords mention cachexia but not sarcopenia. Cachexia is mentioned nowhere within the manuscript.
We want to thank the reviewer for carefully reading the manuscript. We have adapted the keywords as suggested.
Introduction
There is a poorly defined and increasingly murky relationship between cachexia and sarcopenia. There is likely to be a large degree of overlap and/or, both conditions being syndromes within the same spectrum of pathophysiology. However, cachexia is an accepted term when considering muscle loss in cancer and therefore it should be introduced within the background information with a brief discussion about terminology. In particular considering why sarcopenia has been chosen as a term for this study especially given that measures of muscle strength and performance have not been included.
We want to thank the reviewer for this important comment. To our best knowledge, sarcopenia is defined as “the progressive loss of muscle mass and strength with a risk of adverse outcomes such as disability, poor quality of life and death” by the Special Interest Group of the European Sarcopenia Working Group in 2010. Cachexia is defined as a metabolic syndrome in which inflammation is the key feature and so cachexia can be an underlying condition of sarcopenia. Since our study does not feature much data needed to fulfill the criteria for cachexia, we believe that sarcopenia better reflects what we have analyzed.
Methods
L3SMI formula appears to be referenced wrong.
We would like to thank this reviewer for her/his careful evaluation of our manuscript. The reference has been updated.
There is some variation in the literature about whether L3SMI includes adjustment for height or height squared. Please reference appropriately.
We would like to thank this reviewer for this important comment. There is indeed some variation in the literature about the adjustment of L3SMI with respect to patients high. We have chosen to adjust for the patients’ height. Importantly, we would like to emphasize that the results of our analyses do not depend on this decision. As such, the Cox-regression analysis for the prediction of OS show as HR of 0.877 (95%CI: 0.827-0.930, p<0.001) for the L3SMI adjusted for height square, which is comparable to our presented results (HR: 0.929, 95%CI: 0.898-0.960, p<0.001). To fully comply with this important comment, we have added a respective section on this important issue in the revised discussion section of our manuscript as follows:
“In addition, there is some variation in the literature about methods of adjustments regarding the L3SMI [9, 28–30], which should be taken into account when comparing results between studies.” (page 11 of the revised manuscript)
I was unable to find the length of follow up in the methodology which is important given the quotation of survival statistics later on.
We would like to thank the referee for this important statement and fully agree that the median length of follow-up is essential for the reader to interpret the survival data. We have now included this highly relevant information in the revised materials and methods section as follows:
“The median time of follow-up (time between first administration of ICI therapy and death/censoring) was 257 days (IQR: 127-526 days).” (page 4 of the revised manuscript)
I would also mention the multivariable analysis within the statistical methods.
This is again a very relevant statement, which we would like to thank this referee for. We have added a respective paragraph in the multivariate Cox-regression analysis in the revised materials and method section of our manuscript:
“The prognostic value of variables was further tested in uni- and multivariate Cox re-gression analyses. Parameters with a p-value <0.100 in univariate analyses were in-cluded into the multivariate Cox-regression analysis. The hazard ratio (HR) and the 95% confidence interval are displayed.” (page 4 of the revised manuscript)
Results
There is no mention of ethnicities in the patient characteristics. This can have a significant impact on body composition. This needs to be either included or considered as a limitation.
This is an important reviewers´ comment. We want to emphasize that our institution does not systematically record the ethnicity of the patients treated, so we cannot make any statement on this. As suggested by the reviewer, we have now discussed this point in the limitations.
“Moreover, our database did not feature information regarding ethnicities, which potentially might have an influence on body composition.” (page 10 of the revised manuscript).
I would also like to see baseline L3SMI and MMA analysed with respect to age as this is an important factor within the development of sarcopenia.
We want to thank the reviewer for this interesting comment. We have now performed a correlation analysis between baseline L3SMI/MMA values and patients’ age. Interestingly, we did not observe a significant correlation between the L3SMI and the patients’ age in our cohort of patients (rS: 0.066, p=0.546). However, the MMA showed a negative correlation with patients’ age as suggested by this reviewer. We have included this interesting novel data into the revised result section of our manuscript as shown below. We would like to emphasize that patients’ age was not associated with OS in univariate Cox-regression analysis (see Table 2).
“The L3SMI did not correlate with patients’ age (rS: 0.066, p=0.546).” (page 5 of the revised manuscript).
“The MMA negatively correlated with patients’ age (rS: -0.318, p=0.003).” (page 5 of the revised manuscript).
You state that you identified patients with increasing and decreasing L3SMI and MMA. Reviewing the figure most of the patients appear to be static so the actual change would be very small and of little clinical relevance. I am not really sure this statement adds anything to the manuscript at all. I would probably remove it or clarify why you think it is important.
We agree with the reviewer that the figure might be misleading. We have removed the respective information from the manuscript as kindly suggested.
I would like to see the full univariable Cox analysis for MMA. Maybe in the supplementary data.
As requested by this reviewer, we have added the univariable Cox-regression analysis for MMA. We agree with this reviewer that this analysis is of high relevance. Therefore, we have added this novel data in the new result section of our revised manuscript:
“Univariate Cox regression analysis further revealed a HR of 0.940 (95% CI: 0.881-1.003, p=0.062) for ∆MMA.” (page 8 of the revised manuscript)
Discussion
Air displacement plethysmography is not a frequently used assessment tool of body composition. I would consider ultrasound if you want to quote another example.
We have quoted ultrasound as another tool for assessment of body composition, as kindly suggested. We want to thank the reviewer for this comment.
“The body composition can be analyzed using different methods, with bioelectrical impedance analysis or ultrasound representing the most frequently applied ones [24, 25].” (page 9 of the revised manuscript).
Line 303 - grammatical error/ Line 314 - no reference
We want to thank the reviewer for carefully reading the manuscript. The requested corrections have been done.
I would really emphasise the survival data in the discussion eg the difference between 127 and 547 days. This is huge and clinically very important both for clinicians and patients. In my opinion what this paper has highlighted is 1) an excellent prognostic tool and 2) possible mechanisms for sarcopenia/muscle loss. Although point 2 is discussed I would try and emphasise both these points more in the discussion.
We are grateful to the reviewer for her/his positive feed-back. Nevertheless, we believe that it is important not to overstate our results, considering the limitations of our study (please see new paragraph within the discussion section of the revised manuscript). We are currently preparing multi-centric prospective studies further analyzing the role of sarcopenia and myosteatosis in cancer patients receiving ICI to enhance evidence for a potential use of both parameters as prognostic tools in clinical routine. Again, we want to thank the reviewer for this comment.
As a limitation to the methodology I would include the liklihood of concurrent multi-morbidity that could increase prevalence of sarcopenia with certain cancers. Eg NSCLC and COPD both of which can cause cachexia and have independent inflammatory effects.
We fully agree with the reviewer on this point. To provide full transparency, we have now included the following sentence into the new discussion section of the revised version of the manuscript:
“Finally, distinct comorbidities including COPD that are more frequent in specific subgroup of patients might have biased the results.” (page 10 of the revised manuscript).